# Effects of 17α-Methyltestosterone on the Transcriptome and Sex Hormones in the Brain of *Gobiocypris rarus*

**DOI:** 10.3390/ijms24043571

**Published:** 2023-02-10

**Authors:** Shaozhen Liu, Yue Chen, Tongyao Li, Liying Qiao, Qiong Yang, Weiya Rong, Qing Liu, Weiwei Wang, Jing Song, Xianzong Wang, Yu Liu

**Affiliations:** College of Animal Science, Shanxi Agricultural University, Jinzhong 030801, China

**Keywords:** 17α-methyltestosterone, rare minnow, RNA-seq, gene expression, brain

## Abstract

17α-Methyltestosterone (MT), a synthetic environmental endocrine disruptor with androgenic effects, has been shown to disrupt the reproductive system and inhibit germ cell maturation in *Gobiocypris rarus*. To further investigate the regulation of gonadal development by MT through the hypothalamic-pituitary-gonadal (HPG) axis, *G. rarus* were exposed to 0, 25, 50, and 100 ng/L of MT for 7, 14, and 21 days. We analyzed its biological indicators, gonadotropin-releasing hormone (GnRH), gonadotropins, reproduction-related gene expression, and brain tissue transcriptome profiles. We found a significant decrease in the gonadosomatic index (GSI) in *G. rarus* males exposed to MT for 21 days compared to the control group. GnRH, follicle-stimulating hormone (FSH), and luteinizing hormone (LH) levels, as well as the expressions of the *gnrh3*, *gnrhr1*, *gnrhr3*, *fshβ*, and *cyp19a1b* genes, were significantly reduced in the brains of both male and female fish when exposed to 100 ng/L MT for 14 days compared to the controls. Therefore, we further constructed four RNA-seq libraries from 100 ng/L MT-treated groups of male and female fish, obtaining 2412 and 2509 DEGs in male and female brain tissue, respectively. Three common pathways were observed to be affected in both sexes after exposure to MT, namely, nicotinate and nicotinamide metabolism, focal adhesion, and cell adhesion molecules. Furthermore, we found that MT affected the PI3K/Akt/FoxO3a signaling pathway through the upregulation of *foxo3* and *ccnd2*, and the downregulation of *pik3c3* and *ccnd1*. Therefore, we hypothesize that MT interferes with the levels of gonadotropin-releasing hormone (GnRH, FSH, and LH) in *G. rarus* brains through the PI3K/Akt/FoxO3a signaling pathway, and affects the expression of key genes in the hormone production pathway (*gnrh3*, *gnrhr1* and *cyp19a1b*) to interfere with the stability of the HPG axis, thus leading to abnormal gonadal development. This study provides a multidimensional perspective on the damaging effects of MT on fish and confirms that *G. rarus* is a suitable model animal for aquatic toxicology.

## 1. Introduction

In Asia, the production and consumption of chemicals are increasing at an unprecedented rate; hence, there is a need for their management [1]. The aquatic environment is the habitat on which aquatic organisms, such as fish, depend, and most environmental pollutants eventually accumulate in the aquatic environment. Thus, pollutants pose a serious threat to the growth and development of fish. Fish are a representative group of aquatic organisms that are sensitive to changes in the aquatic environment, and are, therefore, often used to monitor environmental and aquatic ecotoxicology studies [2]. *Gobiocypris rarus* is a small fish that is endemic to China. It is mainly found in the upper reaches of the Yangtze River, in tributaries of the Dadu River and in small rivers near Chengdu, Sichuan [3]. *G. rarus* has a small body size, a fast reproductive cycle, a high fertilization rate and hatching rate, and controllable embryonic development. It is widely used in studies of ichthyology, genetics, embryology, physiology, and ecotoxicology [4,5,6,7]. It also has great potential for water quality assessment [8]. The sensitivity of zebrafish (*Danio rerio*), fathead minnow (*Pimephales promelas*), and Chinese rare minnow (*G. rarus*) to various chemicals was systematically investigated using chemical proportional and probabilistic chemical toxicity distributions. The results showed that *G. rarus* was more sensitive than *D. rerio* and *P. promelas*, especially to heavy metals [1].

17α-Methyltestosterone (MT) is a synthetic androgen. It is usually found in industrial wastewater, domestic sewage and aquaculture wastewater. In 2011, 250 ng/L MT was detected in waters near a wastewater treatment plant in the United States, and 4.1~7.0 ng/L MT was detected in urban area wastewater in Beijing in 2010 [9,10]. Androgens play an important role in sexual maturation, gonadal differentiation, spermatogenesis, the development of secondary sexual characteristics, and reproduction in male bony fish [11]. They also play an important role in female fish oogenesis; androgen receptors are present in the ovaries, and androgens act not only as controllers of oocyte development, but also as estrogen precursors for aromatase conversion in many bony fish [12]. Androgens can also bind directly to estrogen receptors [13]. MT was studied in the 1970s, and different fish species responded differently to MT [14]; however, the effects of MT on fish were not absolutely positively or negatively correlated [15,16]. MT exposure can promote yolk production by increasing follicle diameter and serum 11-ketotestosterone (11-KT) levels in Japanese eels [17]. It inhibits gonadal development, interferes with the expression of genes that are critical for gonadal development, and inhibits the synthesis of sex steroids in *Pseudorasbora parva* [18].

It is well known that fish reproduction is controlled by the hypothalamic-pituitary-gonadal (HPG) axis, and MT can interfere with the endocrine system when it enters the fish. The central parts of the endocrine system in fish are the hypothalamus and pituitary gland, which respond to neural signals from the brain and translate them into chemical signals. MT can significantly induce ovarian development, increase oocyte diameter, and stimulate ovarian maturation in eels [17]. MT exposure causes abdominal swelling and reduced fertility in female *Oryzias latipes* during the breeding season, and can also cause females to develop androgenic traits [19]. MT exposure can cause significant increases in VTG in juvenile *Carassius auratus* and *P. promelas* [20,21]. Previous studies have shown that MT can affect gonad histology and gene expression in fish gonads [22]. We hypothesize that MT influences gonadal development of *G. rarus* by regulating genes upstream of the HPG axis. Therefore, we aimed to investigate the effects of MT on the upstream of the HPG axis, and to uncover the mechanism of MT action in *G. rarus*.

Therefore, in this study, *G. rarus* was exposed to MT at different concentrations (25–100 ng/L) for 7, 14, and 21 days (d), in order to observe the biological parameters, gonadotropin-releasing hormone (GnRH), follicle-stimulating hormone (FSH), and luteinizing hormone (LH) levels in the brain, as well as the expression levels of genes related to reproductive regulation. Brain tissue from male and female fish that were exposed for 14 days in the control and 100 ng/L MT groups was selected for transcriptome sequencing.

## 2. Results

### 2.1. Effects of MT on Biological Parameters and Gonadal Indices in G. rarus

In females, fish full length, body length, and body weight significantly decreased after exposure to 25 ng/L of MT for 7 days. GSI significantly increased after exposure to 50 ng/L of MT for 7 days. The weight of the fish significantly increased after exposure to 50 and 100 ng/L of MT for 21 days. In males, full length and body weight were significantly higher after exposure to 50 ng/L of MT for 7 days (Figure 1).

### 2.2. Effects of MT on Reproduction-Related Hormone Content in the Brain of G. rarus

In females, GnRH, LH, and FSH in the brain significantly increased after exposure to 25 ng/L of MT for 7 days, and decreased gradually with increasing exposure. GnRH, LH, and FSH levels decreased significantly after exposure to 25 and 100 ng/L of MT for 14 days. GnRH, LH, and FSH significantly declined after exposure to 50 ng/L of MT for 21 days. In males, no significant changes in all three hormone levels were observed after 7 and 21 days of 0, 25, 50, and 100 ng/L MT exposure. GnRH, LH, and FSH were significantly reduced after exposure to 50 and 100 ng/L of MT for 14 days (Figure 2).

### 2.3. Effects of MT on the Expressions of Genes Related to Reproduction in G. rarus

In female fish, *cyp19a1b* expression was significantly reduced after exposure to 25 ng/L of MT for 7 days, and to 50 and 100 ng/L of MT for 14 days; *cyp19a1b* expression significantly increased for 21 days of exposure to all three concentrations (25, 50, and 100 ng/L) of MT. Exposure to 25, 50, and 100 ng/L of MT for 7 days resulted in a significant decrease in *fshβ* expression. The expression of *fshβ* significantly decreased in both groups that were exposed to 100 ng/L of MT for 14 days, and to 25 and 50 ng/L of MT for 21 days. The expression of *lhβ* was significantly reduced when exposed to all three concentrations of MT for 7 days. The expression of *lhβ* significantly increased by exposure to 50 ng/L of MT after 14 days, and by exposure to 25 and 50 ng/L of MT for 21 days. *lhβ* expression significantly increased when exposed to 50 ng/L of MT for 14 days, and to 25 and 50 ng/L of MT for 21 days. Exposure to three concentrations of MT for 7 and 14 days significantly increased the expression of *gnrh2* in *G. rarus*. The expression of *gnrh2* was significantly elevated with exposure to 25 and 100 ng/L of MT for 21 days. Exposure to all three concentrations of MT for 7 and 14 days significantly decreased *gnrh3* expression, and when extended to 21 days, *gnrh3* expression significantly increased in the group exposed to 50 ng/L of MT. The expression of *gnrh1* was significantly reduced in all three MT concentrations for 7 and 14 days, and the expression of *gnrh3* significantly increased when exposed to 25 and 50 ng/L of MT for 21 days. The expression of *gnrhr3* significantly increased after exposure to 50 ng/L of MT for 7 days. *gnrhr*3 expression significantly decreased in all three concentrations of MT when extended to 14 days. The expression of *ar* was significantly elevated after exposure to 25 ng/L of MT for 21 days. The expression of *erα* was significantly elevated both at 50 ng/L of MT for 14 days, and at 25 ng/L of MT for 21 days. The expression of *erβ* was significantly higher for all three concentrations of MT exposure for 21 days (Figure 3).

In male fish, the expression of *gnrh3* significantly decreased when exposed to 50 ng/L of MT for 14 days, and to all three concentrations of MT for 21 days. The expression of *gnrhr1* significantly decreased when exposed to 25 and 50 ng/L of MT for 7 days, and in the three concentrations of MT for 14 days; when extended to 21 days, the expression of *gnrhr1* was significantly higher when exposed to 50 ng/L of MT. The expression of *gnrhr3* was significantly higher when exposed to 25 ng/L of MT for 7 days. The expression of *gnrhr3* was significantly higher in the three concentrations of MT when extended to 14 days. The expression of *gnrhr3* was significantly reduced in all three concentrations of MT. The expression of *fshβ* significantly increased both when exposed to 100 ng/L of MT for 14 days, and to 50 ng/L of MT for 21 days. The expression of *lhβ* was significantly elevated when exposed to all three concentrations of MT for 7 days. The expression of *lhβ* was significantly lower when exposed to all three concentrations of MT when extended to 14 and 21 days. The expression of *cyp19a1b* significantly decreased when exposed to 100 ng/L of MT for 7 days, and to 25 ng/L of MT for 14 days. The expression of the *erβ* gene significantly increased when exposed to all three concentrations of MT for 7 days, and when the exposure was extended to 14 days, the expression of *erα* became significantly higher when exposed to 50 ng/L of MT. The expression of the *erβ* gene was significantly higher when exposed to 25 ng/L of MT for 7 days, and to 100 ng/L of MT for 14 days (Figure 4).

### 2.4. Changes in the Brain Transcriptome following MT Exposure

#### 2.4.1. Sequencing Evaluation

In the present study, the number of clean reads in the male and female control groups were 20,333,009 and 20,364,461, respectively, while the number of clean reads in the 100 ng/L MT-treated group was 20,781,655 and 20,490,045 for males and females, respectively. There were 99,929 Unigenes obtained after assembly, with an N50 of 3335.

#### 2.4.2. Screening for DEGs

A total of 2412 differentially expressed genes (DEGs) were screened in female fish brains after MT exposure, of which 1370 were upregulated and 1042 were downregulated, and a total of 2509 DEGs were screened in male fish brains, of which 1079 were upregulated and 1430 were downregulated (Figure 5).

#### 2.4.3. Functional Notes for DEGs

A total of 52 Gene Ontology (GO) items, including 22 biological processes, 17 cellular components, and 13 molecular functions, were significantly enriched in the brain tissue of female *G. rarus* after MT exposure. In contrast, a total of 51 GO items, including 22 biological processes, 17 cellular components, and 12 molecular functions, were significantly enriched in male *G. rarus* (Figure 6).

After MT exposure, the Kyoto Encyclopedia of Genes and Genomes (KEGG) pathway in female fish involved five major categories, namely cellular processes, environmental information processing, genetic information processing, metabolism, and organismal systems, with the largest proportion of genes in the pathway being focal adhesion in cellular processes, with a total of 10 genes. In addition to these five pathways, male fish are also involved in human diseases, and the pathway with the largest proportion of genes is Salmonella infection in human diseases, with a total of 10 genes (Figure 7).

After MT exposure, in females, the 20 most significant KEGG pathways were enriched, with increased expression in one KEGG pathway, the Notch signaling pathway, and decreased expression in two KEGG pathways, namely biotin metabolism and the phosphatidylinositol signaling system. In males, the 20 most significantly enriched KEGG pathways were those in which increased expression was found in only one KEGG pathway, oxidative phosphorylation, and decreased expression was found in three KEGG pathways. Three pathways were significantly enriched in both sexes, namely nicotinate and nicotinamide metabolism, focal adhesion, and cell adhesion molecules (Figure 8).

#### 2.4.4. qRT-PCR to Verify the DEGs

The RNA-Seq results of each of the eight genes in the brains of male and female *G. rarus* were validated with qRT-PCR. In the qRT-PCR results with RNA-Seq results, it was observed that in females, *cry1*, *foxo3*, and *ccnd2* were upregulated, and *pms2*, *wtap*, and *pik3c3* were downregulated. Meanwhile, in males, *tpr*, *cpd*, and *arntl* were upregulated, and *fundc1*, *rbbp4*, and *etv4* were downregulated (Table 1). The results of the two methods are consistent, and can prove that the current transcriptome sequencing data are reliable.

## 3. Discussion

Environmental endocrine disruptors (EEDs) can compete with sex hormone receptors and prevent endogenous sex hormones from binding to the receptors for action, thus affecting the normal physiological functions of sex hormones [23]. In this study, we found that the expression of GSI significantly decreased in female *G. rarus* exposed to 25 and 50 ng/L of MT for 14 days. The expression of GSI significantly decreased when exposed to three concentrations of MT (25, 50, and 100 ng/L) for 21 days. Previous studies by our group showed that gonadal degradation was progressively more severe, and that the proportion of mature oocytes and mature sperm decreased with increasing exposure time and exposure concentration [24]. In addition, MT can inhibit gonadal development in *Epinephelus coioides* [25], *Anguilla japonica* [17], *Anoplopoma fimbria* [26], *Kryptolebias marmoratus* [27], *Pelteobagrus fulvidraco* [28], and *P. parva* [18]; these results suggest that MT affects spermatozoa and ovary development, and hinders the maturation of germ cells in fish.

MT can affect the gonadotropin and sex hormone levels in *A*. *japonica* [17], *Centropomus undecimalis* [29], *P. parva* [18], and *E*. *coioides* [30]. The results of this study showed that MT disrupts the levels of GnRH and gonadotropins (LH and FSH); all three hormones increased and then decreased with increasing concentrations in female *G*. *rarus* exposed to MT for 7 days. When exposed to MT for extended periods of up to 21 days, all three hormones first decreased with increasing exposure concentrations, and then returned to normal levels. This may be due to the physiological compensatory ability of the organism to adapt to different environmental states within a certain physiological range [31]. At the same time, we found that changes in hormone content of female fish are higher than that of male fish. We speculate that female hormones are more sensitive to MT, which is consistent with previous results [18]. *gnrh2*, *gnrh3*, *gnrhr1*, and *gnrhr3*, which are in *G. rarus*, were expressed mainly in brain tissue [32,33]. In males, none of the *gnrh2* expressions significantly changed, while the expression of *gnrh3* decreased when exposed to different concentrations of MT after 14 and 21 days, suggesting that *gnrh2* may not be involved in the regulation of the HPG axis induced by MT exposure. This result is the same for other species, such as bullfrogs, chickens, and the musk shrew (*Suncus murinus*) [34]. MT may interfere with the GnRH system by altering the number of *gnrh3* neurons, a result that was identical to that of *D. rerio* treated with 29.6 ng/L EE2 [35,36]; this further demonstrated that androgens act through conversion to estrogens. Exposure to 50 ng/L MT for 21 days led to elevated expressions of *gnrhr1* and *gnrhr3* in the brain, with 16.3 times more *gnrhr1* expression and 3.3 times more *gnrhr3* expression than the control. This indicates that *gnrhr1* is more sensitive to MT’s transcriptional regulation than *gnrhr3* in the *G. rarus* brain. GnRH secreted by the hypothalamus stimulates the release of pituitary FSH and LH, which regulate gonadal steroids and, thus, endocrine regulation through positive and negative feedback [37,38]. The expression of *gnrh3* in the brain was significantly reduced in both females that were exposed to different concentrations of MT for 7 days, and in males exposed to different concentrations of MT for 21 days, which is the same expression pattern of *lhβ* in the brain, indicating the possibility that *gnrh3* is the main isoform of GnRH that exerts a pro-pituitary function [39]. We, therefore, hypothesize that MT interferes with the KISS/GPR54 system through the *gnrh*3 gene, thus interfering with gonadal development [40]. Cytochrome P450 aromatase (*cyp19a1*) is a rate-limiting enzyme in estrogen synthesis that catalyzes the conversion of androgens to estrogens. *cyp19a1b* is mainly expressed in brain tissue [41]. In the present study, *cyp19a1b* in the brain showed different trends in the MT-treated group, indicating a disruption of steroid homeostasis. The results of the present study showed that MT treatment disrupts the levels of GnRH, LH, and FSH, leading to disruption of HPG axis-related genes (*gnrh2*, *gnrh3*, *gnrhr1*, *gnrhr3*, *fshb*, and *lhβ*), steroid synthesis-related genes (*cyp19a1b*), and androgen receptor genes (*ar*, *erα*, and *erβ*), and their expression patterns. Therefore, we suggest that MT may affect hormone levels in the brain by interfering with the expressions of key genes in the hormone production pathway, thereby interfering with the stability of the HPG axis.

In this study, a male *G. rarus* exposed to 100 ng/L MT for 14 days had a total of 52 GO items, including 22 biological processes, 17 cellular components, and 12 molecular functions, that were significantly enriched in brain tissue. In GO entries, DEGs are mainly enriched for biological processes, such as bioregulation, metabolic processes, and stress responses; for molecular components, such as cell components, organelles, and cell membrane components; and for molecular functions, such as catalytic activity, transport dynamics, molecular functional regulators, and signaling activities. Among them, females are more enriched than males in one molecular function, metallochaperone activity. The KEGG enrichment analysis revealed that DEGs in female fish were enriched in 200 signaling pathways, of which the most significant 20 pathways had increased expression, including the Notch signaling pathway. The Notch signaling pathway is a highly evolutionarily conserved signaling pathway that can have important effects on cell proliferation, differentiation, development, migration, apoptosis, and other processes. Moreover, it has a regulatory role in tissue homeostasis and internal environmental stability [42]. Notch is a signaling pathway that is involved in the regulation of the immune response, and can play an important regulatory function in the development of bone marrow dendritic cells, and in the differentiation of helper T lymphocytes. In addition, it regulates several complex processes of T cell and B cell development in the central and peripheral lymphoid organs in the organism. Furthermore, it promotes the immune function of natural killer cells [43,44]. Decreased expression was enriched in two KEGG pathways, the biotin metabolism and phosphatidylinositol signaling systems. DEGs in female fish were enriched in 109 signaling pathways, of which the most significant 20 pathways with increased expression were enriched in the oxidative phosphorylation signaling pathway, and expression decreased in three KEGG pathways. It showed that MT interfered more significantly in males than in females, which is consistent with the results of MT treatment in *P. parva* [18].

During the stage of primordial follicle to primary follicle development, the PI3K/Akt signaling pathway plays a major role in accelerating the recruitment of primordial follicles [45]. PI3K is an important intracellular signaling-inducing molecule that recruits downstream AKT to the cell membrane to promote follicle survival, growth, and oocyte activation by transducing signals [46]. It was found that the expression of *pik3c3* in the PI3K/Akt signaling pathway significantly decreased in the brains of female *G. rarus* after exposure to MT. One of the key downstream transcription factors regulated by the PI3K/Akt signaling pathway is the FoxO protein, and the PI3K/Akt signaling pathway mainly negatively regulates the transcription of the FoxO1, FoxO3a, and FoxO4 proteins. In our study, the expression of *foxo3* was significantly elevated. Studies show that the PI3K/Akt signaling pathway is abnormally activated in ovarian cancer tissues compared to normal tissues, with higher PI3K mutations and increased Akt expression, and the PI3K/Akt signaling pathway is considered to be one of the major pathways involved in ovarian progression [47,48]. Treatment of rats with the organochlorine pesticide pentachloronitrobenzene activated the PI3K/Akt signaling pathway and accelerated the development of their primordial follicles [49]. Our study shows that exposure to MT in *G. rarus* will lead to abnormalities in their gonadal development, which may be related to MT regulation of the PI3K/Akt/FoxO3a signaling pathway.

## 4. Materials and Methods

### 4.1. Experimental Animals

This study was approved by the Shanxi Agricultural University Animal Care and Ethical Committee, China, the IACUC No. is SXAU-EAW-2022F.BN.001012001. During the experiment, fish were humanely treated. *G. rarus* (♀:♂ = 1:1, 8 months) were selected from the laboratory of the Department of Aquatic Sciences, Shanxi Agricultural University. Four groups were set up in the experiment; the control group was 0.001% anhydrous ethanol, and the treatment group was *G. rarus* exposed to MT (25, 50, and 100 ng/L), with males and females reared separately. Three parallel replicas were set up for each test group. A total of 24 aquariums (80 L volume, 60 L water) were used, with 30 *G. rarus* in each aquarium. The water used for this experiment was tap water after 24 h of aeration and semi-clean water exposure, with the pH range 7.6 ± 0.2, and a temperature range of 25 ± 1 °C; the photoperiod was artificially controlled at 14 h: 10 h light: dark. The stocking density was 1 g of fish per 1 L of water; half of the water was changed in the aquarium while sucking out the effluent (residual bait and feces), adding the same amount of water, and the corresponding amount of MT solution at the same time; this was carried out to ensure that the MT concentration in the aquarium remained unchanged. The fish were fed red worms once a day at regular intervals, at a feeding rate of 3% of the total weight of the fish in the experimental group; the health status of the test fish in each group was observed and recorded. Feeding was stopped the day before sampling, and 18 *G. rarus* were randomly selected from each group, and brain and gonadal tissues were rapidly removed after anesthesia with MS222.

### 4.2. Measurement of Biological Indicators

The biological indicators measured included the full length, body length, body weight of the fish. The gonadosomatic index (GSI = gonad weight/body weight × 100%) was calculated from the gonad weight and body weight.

### 4.3. ELISA

According to previously published methods [50], the brain tissue of the sample was removed (*n* = 3 per group) and immediately transferred to a heparinized centrifuge tube. Protease inhibitor (2 trypsin inhibitor units/mL) was added to the tubes, which were then centrifuged at 21,380× *g* for 15 min at 4 °C. The supernatant was carefully pipetted out and stored at −80 °C for further determination of gonadotropin-releasing hormone (GnRH), follicle stimulating hormone (FSH), and luteinizing hormone (LH) levels, performed using commercial ELISA kits (Nanjing Jiancheng Biotechnology Co., Ltd., Nanjing, China), according to the manufacturer’s protocols. All samples and standards were replicated three times (both inter-assay and intra-assay CV were less than 10%).

### 4.4. RNA Sample Collection and Quality Testing

*G. rarus* brain tissues were stored in an ultra-low-temperature refrigerator. Total RNA was extracted using the Trizol one-step method, RNA integrity was detected via 1% agarose gel electrophoresis, and purity and concentration were determined spectrophotometrically. Each tissue sample was taken in 5 μL amounts, and the cDNA first strand was reverse transcribed using the PrimeScriptTM RT Reagent Kit with gDNA Eraser (Perfect Real Time).

### 4.5. qRT-PCR

The relative expressions of *cyp19a1b*, *fshβ*, *lhβ*, *gnrh2*, *gnrh3*, *gnrhr1*, *gnrhr3*, *ar*, *erα*, and *erβ* genes in brains were measured using qRT-PCR, with specific primers as shown in Table 2 [51,52]. The TB Premix Ex TaqTM II (Tli RNase H Plus) kit (TaKaRa, Dalian, China) was used for the experiments, and three replicates were set up for each tissue treatment. The average threshold cycle (Ct) was used for calculating expression level by the 2^−ΔΔCt^ method.

### 4.6. Sample Sequencing

#### 4.6.1. RNA Preparation and Sequencing

In this study, female fish brain tissue from the control group (CON-F), male fish brain tissue from the control group (CON-M), female fish brain tissue exposed to MT100 (MT100-F), and male fish brain tissue exposed to MT100 (MT100-M) were selected for sequencing in the treatment group. A total of 12 samples were used, with 3 replicates in each group (*n* = 12 groups in total, with *n* = 3 per group), which were used for library construction and sequencing by Bemac Biotechnology Co., Tokyo, Japan.

#### 4.6.2. Bioinformatics Analysis

Raw data were filtered to remove splice sequences and low-quality reads to obtain high-quality clean data, which were then assembled to obtain Unigene libraries. Randomness and saturation tests were performed to assess the quality of the sequencing libraries. Expression analysis, gene structure analysis, differential expression analysis, functional annotation of differentially expressed genes (GO, COG, KEGG), and functional enrichment (GO, KEGG) analysis were performed based on the expression of genes in different samples or different sample groups. We identified DEGs with a fold change ≥ 2 and a false discovery rate (FDR) ≤ 0.05 from comparisons across samples or groups.

#### 4.6.3. RT-qPCR Analysis for Validating Expression Profiles of Transcriptome Sequencing

Eight genes were selected in the male and female groups, and qRT-PCR was performed to validate the RNA-Seq data. The specific primers used for qRT-PCR validation are shown in Table 1. Six RNAs from each group (*n* = 6 per group) were used for qPCR validation. The average threshold cycle (Ct) was used for calculating expression level by the 2^−ΔΔCt^ method.

### 4.7. Statistical Analysis

All data were analyzed by SPSS 21.0 (IBM Inc., Chicago, IL, USA) and presented as mean ± SD. One-way ANOVA with Dunn’s post hoc test was used. *p* < 0.05 indicated a significant difference. Statistically significant differences are indicated by different letters.

## 5. Conclusions

In the present study, we found that (25–100 ng/L) MT inhibits gonadal development and hinders germ cell maturation in *G. rarus*. MT interferes with the levels of GnRH and gonadotropins in the brain, and affects the expression of key genes in the hormone production pathway, interfering with the stability of the HPG axis. To better understand the interference of MT in the brain tissue of *G. rarus*, we constructed RNA-seq libraries of four of them, and exposed males and females to 100 ng/L of MT for 14 d, obtaining 2412 and 2509 DEGs in male and female brain tissue, respectively. After exposure to MT, there are three common pathways in both sexes, namely, nicotinate and nicotinamide metabolism, focal adhesion, and cell adhesion molecules. We found that exposure of *G. rarus* to MT leads to abnormal gonadal development, which may be related to MT regulation of the PI3K/Akt/FoxO3a signaling pathway.

## Figures and Tables

**Figure 1 ijms-24-03571-f001:**
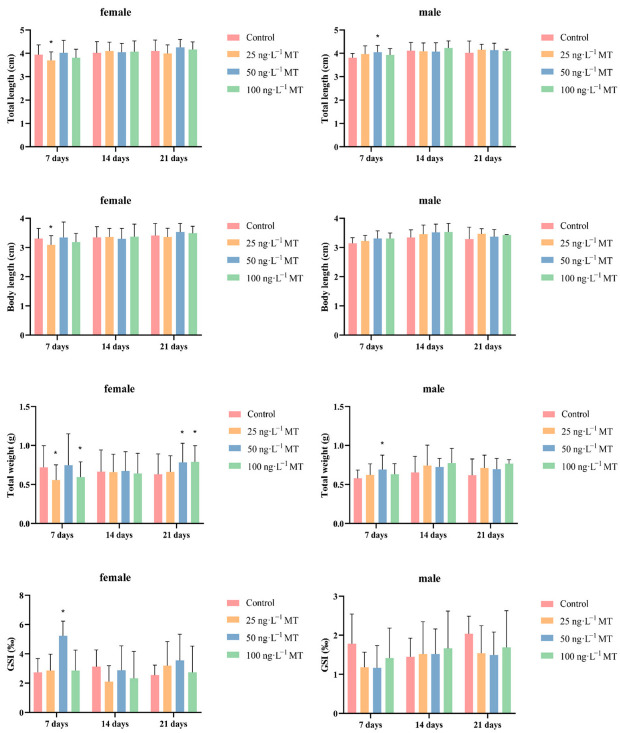
Morphological indexes and gonadosomatic index (GSI) of *Gobiocypris rarus* following MT exposure. Data are expressed as mean ± SD. Results were considered statistically significant for *p* < 0.05 (*).

**Figure 2 ijms-24-03571-f002:**
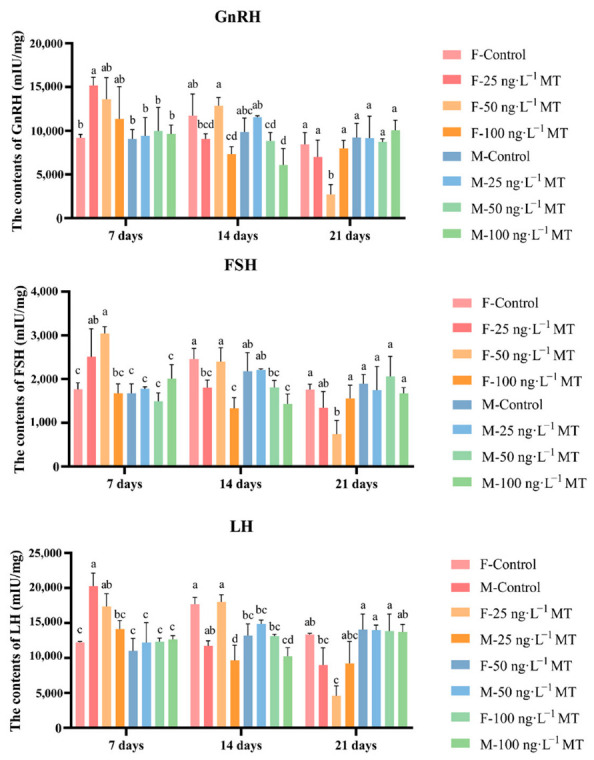
Effects of different doses of 17α-methyltestosterone on the concentrations of GnRH, FSH, and LH in *G. rarus*. Data are expressed as mean ± SD. Statistically significant differences are indicated by different letters (*p* < 0.05).

**Figure 3 ijms-24-03571-f003:**
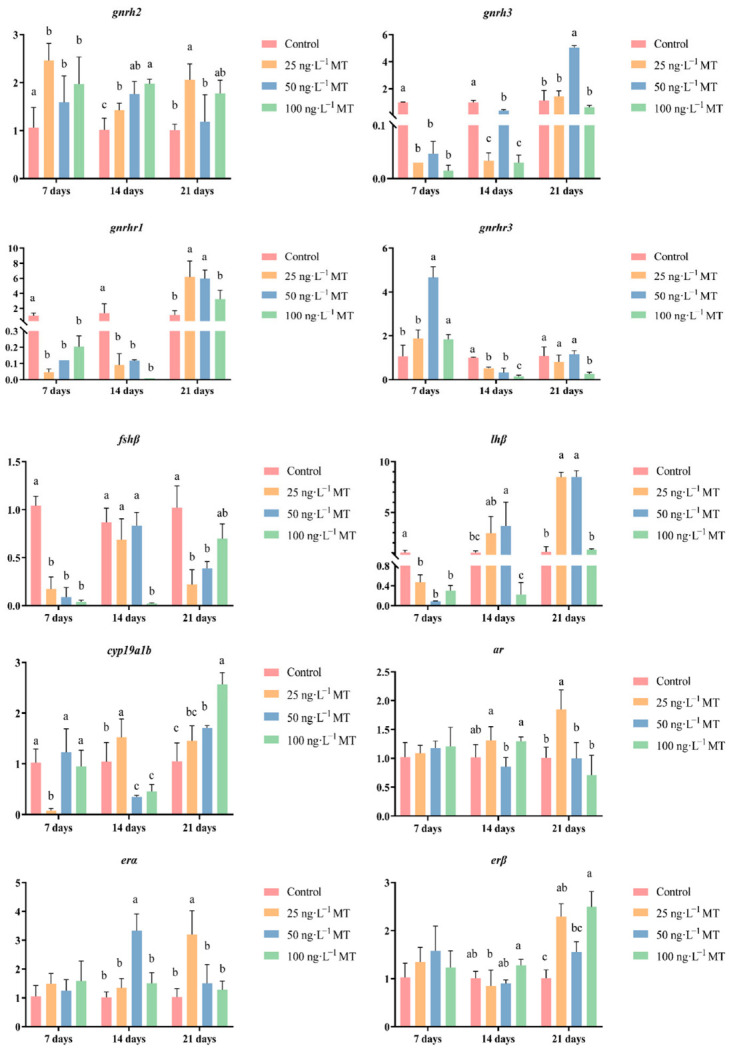
Effect of MT on *gnrh2*, *gnrh3*, *gnrhr1*, *gnrhr3*, *fshβ*, *lhβ*, *cyp19a1b*, *ar*, *erα*, and *erβ* genes in the brain of female *G. rarus*. Data are expressed as mean ± SD. Statistically significant differences are indicated by different letters (*p* < 0.05).

**Figure 4 ijms-24-03571-f004:**
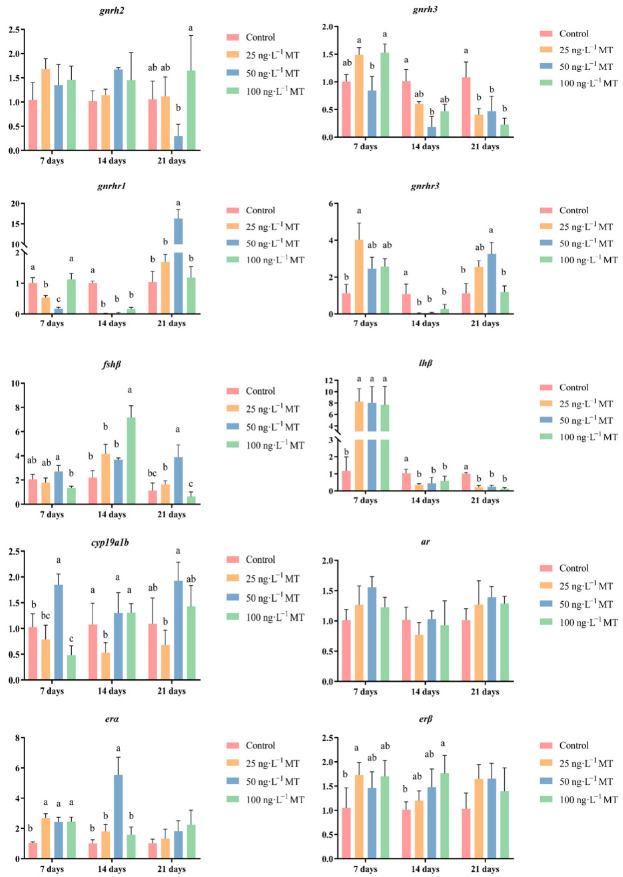
Effect of MT on *gnrh2*, *gnrh3*, *gnrhr1*, *gnrhr3*, *fshβ*, *lhβ*, *cyp19a1b*, *ar*, *erα*, and *erβ* genes in the brain of male *G. rarus*. Data are expressed as mean ± SD. Different letters indicate statistically significant differences (*p* < 0.05).

**Figure 5 ijms-24-03571-f005:**
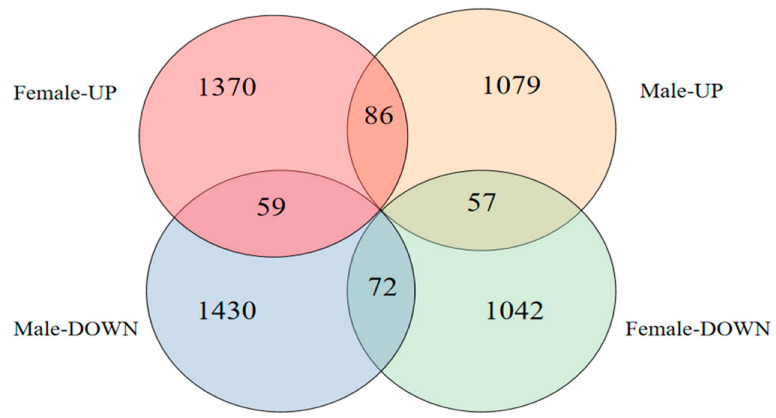
Number of DEGs after MT exposure.

**Figure 6 ijms-24-03571-f006:**
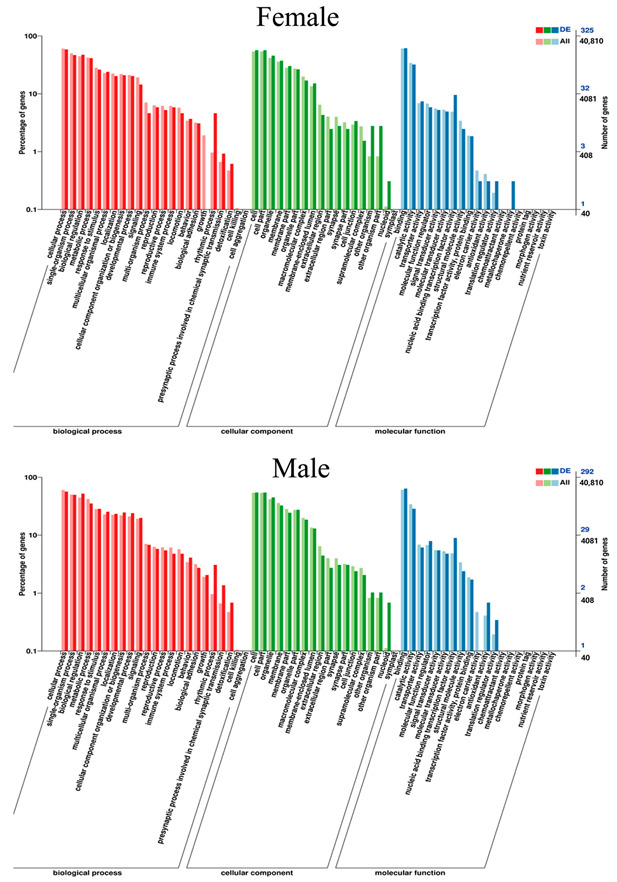
Statistical graph of GO secondary node annotation of differentially expressed genes in *G. rarus* after exposure to MT for 14 days.

**Figure 7 ijms-24-03571-f007:**
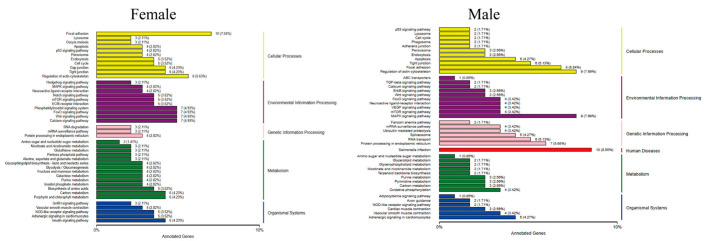
Classification of the differentially expressed gene KEGG in *G. rarus* after 14 days of exposure to 100 ng/L MT.

**Figure 8 ijms-24-03571-f008:**
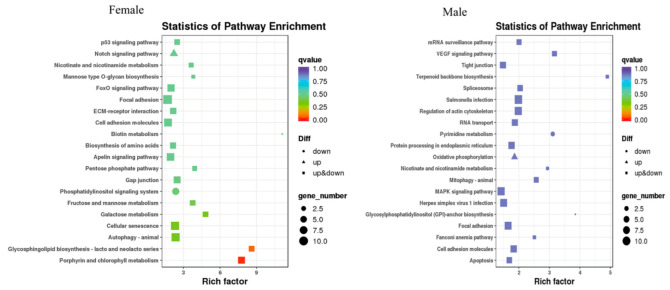
Scatter plot of the differentially expressed gene KEGG pathway in *G. rarus* after 14 days of exposure to 100 ng/L MT.

**Table 1 ijms-24-03571-t001:** Verification of differential expressed genes in *G. rarus* using qRT-PCR. Negative values indicate the down-regulation of expression, whereas positive values indicate the up-regulation of expression.

Sex	Gene	Gene ID	Fold by RNA-Seq	Fold by qRT-PCR
Female	*cry1*	BMK_Unigene_048464	1.74	1.27
	*foxo3*	BMK_Unigene_019235	1.67	1.05
	*ccnd2*	BMK_Unigene_005574	1.53	1.46
	*pms2*	BMK_Unigene_089388	−2.11	−1.32
	*wtap*	BMK_Unigene_183473	−3.50	−1.82
	*pik3c3*	BMK_Unigene_058799	−1.70	−0.13
Male	*tpr*	BMK_Unigene_184367	4.92	0.66
	*cpd*	BMK_Unigene_148720	2.54	0.41
	*arntl*	BMK_Unigene_031409	1.07	0.88
	*fundc1*	BMK_Unigene_095253	−6.40	−1.96
	*rbbp4*	BMK_Unigene_094652	−1.70	−1.81
	*etv4*	BMK_Unigene_098452	−1.44	−0.82

**Table 2 ijms-24-03571-t002:** Primer sequences of qRT-PCR.

Gene	Forward Primer Sequence	Reverse Primer Sequence	Tm (°C)	Primer Length (bp)
*cyp19a1b*	CTAGGTTCCTGTGGATGGG	CGTCGGAGGACTTGCTGA	60	164
*fshβ*	CCATTACCGTGGAAAGTGAGGA	CTGTGATGTCCGAGTTACATTTGC	60	252
*lhβ*	TGAGACTGTTGCTGTGGAAAAAG	AGTATGCGGGGAAATCCTCTC	60	323
*gnrh2*	TGGGGATGTTGCTGTGTCTAAG	TCTGTTTGCTGGAAAGGTCGT	60	285
*gnrh3*	ATGGAGTGGAAAGGAAGGGTG	TGGAGTGTCCGCAGGAATAGA	60	186
*gnrhr1*	GAGCGGAGAGCGAGGACTT	CCAGCAGACCACAAACGACAT	60	284
*gnrhr3*	TGCTCGCCTCTCCACAGTTA	TTCCCCACCCTTCCCTTTAC	60	233
*ar*	CACTGCCAACAAAGGTCAGC	TGGGTGGAGTCGGTATGGAT	60	151
*erα*	TCACCCATGTACCCCAAGGA	GAGTGGTGTCCTCCGTGATG	60	272
*erβ*	AGGGTAGCAGATGCAGAGGA	TCGCCGTAACCCACATTTCA	60	279
*cry1*	TTTTCCCACAATGCACCCTG	GGTTTACTCTCCCATCGCCT	59	160
*foxo3*	CAAAGCCCCTAATGCGATGC	TGGCCGAAAAGTGGAGTTCT	60	166
*ccnd2*	GAGGTGGGTAGGAGGGTCTT	GACAGCCCGCACAAAAGATG	60	219
*pms2*	CACGGCACTACCATCACACT	CTTGCCTTGTCCCATCTGGT	60	153
*wtap*	GACGGGCTCCACTTCGTTTA	GCGAACACTTACCAGACCCA	60	174
*pik3c3*	TTTTCCTCGGGTCTTGTCGG	TGCCACTTCCTTCGGGTTTT	60	102
*tpr*	ATAAAGCCCACACCTCTGGC	ATTGGCTCTTGGCTCTCCAC	60	145
*cpd*	CATCGCATAAACGTCCGCTG	TTACCCAAAGCTCCCGGTTC	60	107
*arntl*	TGGTTTCGGGCAGTATGCTT	CTTCCTCGGCTATCATGCGT	60	170
*fundc1*	CCTCGAAACTGGAGCTTGGT	AGCAGCAGACGGAGTTTTCA	60	116
*rbbp4*	ACCGCTAACAGGGAGGGTTA	CTGGGTCTGCTGAGAATGGG	60	219
*etv4*	AGCTCTTCGTTCTGCCCATC	AATGGTTACAGTGCGGGGAG	60	189

## Data Availability

Not applicable.

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
