# Peer review of "Effects of 17α-Methyltestosterone on the Transcriptome and Sex Hormones in the Brain of Gobiocypris rarus"

_ijms, 2023, doi:10.3390/ijms24043571_

Round 1
Reviewer 1 Report
The authors have submitted a comprehensive and well thought through study. In this study, Liu et al investigated the effect of MT on the regulation of gonadal development through the HPG axis. In short, the authors reported that MT affects gene expression in hormone production pathways and interfere in the brain levels of GnRH and Gonadotropins. They extended their study by sequencing the MT exposed fishes, and reported the significant pathways affected by the differentially expressed genes. The manuscript describes an interesting dataset by studying the effect of MT on the HPG axis of fish. Following are my comments concerning the study.
1. The introduction part was well written providing key information to the readers. However, the source of MT and the acceptable levels of MT along with the levels that were observed in the aquatic water bodies could be included to add depth and significance to this conducted study.
2. For the experiment tap water was used for stocking the fish and treatment in the aquarium. Were the levels of MT tested in the tap water prior to treatment to know the background MT levels? If so, what is the observed level of MT in tap water?
3. Table:2 is difficult to follow with numbers to observe the significant changes. The table could be transformed to a graph for better visibility and appreciate the recorded significant data.
4. Figure-1: What does it mean ‘the contents of GnRH, FSH and LH’ in the Y axis? Also, units are missing. If it is concentration, then the axis title should be changed along with respective units.
5. Figure-1: Sex of the fishes tested was not mentioned on the image.
6. All figures with significant difference lettering: The significant differences are indicated by letters which is very difficult to follow without legends of what those lettering mean. Please explain the letterings in the figure legend to better understand the data.
7. Figure-2 & 3- Since all graphs were from either male or female, instead of the repetition of sex, the gene names can be written on the place of ‘female’ and ‘male’ for better visibility. The sex could be mentioned in the figure legend.
8. Figure 1 & 2- (Recommendation)-There appear to be an observable difference in the hormone levels and gene expression data between male and female sex. It is a recommendation that authors could do a statistic to see if there is any significant sex difference or sex bias upon MT exposure to fishes. It could add more scientific value to the data.
9. Figure 2- How was the data normalized? What is it normalized to? The X axis shows the data is normalized to control but the expression data for all genes shows a bar with fold change for ‘control’ group. Please clarify.
10. Table-3 (Recommendation)- The data could be presented as a Venn diagram showing the commonly expressed genes between sex to understand the proportion of unique DEGs between sex.
11. Figure-4- Y axis title is crammed making it difficult to read.
12. Figure-5- Image resolution is too low to read.
13. In the RNA-seq data, what are the Fold change and significance (p value or FDR) cutoffs used to filter the data?
14. Gonadal development genes blc6, ets1, foxo3, ccnd etc were mentioned to be differentially and enriched in few pathways. The significance of the genes, their level of upregulation or downregulation (Fold change) and why it is important to be seen in canonical pathways and why the author thinks it’s significant need to be discussed. Also please mention the fold difference of genes that are discussed through the study to understand the impact.
15. Figure-7 doesn’t provide much valuable or significant information with very few genes seen in the dataset presented in the pathway.
16. Table-4- Some of the genes vary several folds between RNA-seq and qRT-PCR in the validation with upregulation in RNA-seq and downregulation in qRT-PCR. This variation could either over or under-represent the presented seq data.
17. Last paragraph of discussion- Authors discussed about the role PI3K/AKT pathway played in the ovary based on the expression of two genes pik3c3 and fox03 in the brain. AKT pathway is an important pathway that plays various cell function and regulatory role in various organs. By the expression of 2 genes in brain, the data cannot be extrapolated to the regulation of AKT pathway in the ovary which plays a very important role in the follicle growth and development.

Reviewer 2 Report
International Journal of Molecular Sciences (IJMS-2124573). “Effects of 17α-methyltestosterone on the transcriptome and sex hormones of brain in Gobiocypris rarus”
Reviewer comments:
In general, the present manuscript: “Effects of 17α-methyltestosterone on the transcriptome and sex hormones of brain in Gobiocypris rarus” by Liu and colleagues focused to expose G. rarus to methyltestosterone at different concentrations (25–100 ng/L) for 7, 14, and 21 days (d), in order to observe the biological parameters, gonadotropin-releasing hormone (GnRH), follicle-stimulating hormone (FSH), and luteinizing hormone (LH) levels in the brain, as well as the expression levels of genes related to reproductive regulation, and it has an interesting goal. On the other hand, some specific comments are given bellow, to help improving the quality of the manuscript reviews.
Final remark: this current manuscript needs minor review.
General comments
- In general, I really appreciate this work with careful design and abundant data. In addition, it’s important to note that the manuscript is well written (English grammar and composition). However, the structural organization of sentences needs to be improved, especially in the Results and Discussion sections. Missing description in Material and Methods section was also identified.
Specific comments
Abstract section:
- Conclusion must be made based on the premises described at the final of introduction section. Conclusion strictly related to the results obtained. Depending on the assumptions/suggestions/probability/hypothesis, it can be considered in the discussion. Review the entire manuscript.
- Abbreviation need to be described before they are presented (such as HPG axis).
- Usually, key words are words that do not contain in the title of the manuscript. Review the entire of manuscript.
Introduction section:
- In general, several sentences in the present manuscript are long (mainly in the introduction and discussion). I suggest a reduction so as not to become tiresome to read.
- Currently, most of the scientific manuscript are presented as hypotheses to be more attractive and interesting than description of goals. The present manuscript can be presented with hypothesis. We suggest the authors to present this manuscript with more attractive hypothesis and to make the manuscript more interesting.
Material and Methods:
- The authors mention at the beginning of the M&M description that the present study was approved by the ethics committee (Shanxi Agricultural University Animal Care). However, it is necessary to present the authorization/protocol number. Could the authors provide?
- Still in this first sentence, the authors inform about the replacement of aquarium water to maintain(same) water quality parameters and MT concentration. However, it is necessary to describe some information:
- Has the amount of MT in the water been quantified? Justify.
- Was the amount of MT in the animal tissues quantified? Justify.
- In general, these data are very important to be described in ecotoxicology studies.
- 2.3 ELISA. Provide some information’s:
- Unit in “g” instead of “rpm”;
- For the determination ELISA method. As far as I know, this method has a large degree of error and inaccuracy. The preferred method seems to be HPLC. In addition, the detailed information about the ELISA reagent was not explained clearly, and the source of the reagent antibody was not explained. The steps of hormone determination and serum dosage were not explained. Please supply relevant information. Additionally, validation test (intra-, inter-, and parallelism assays need to be described), working dilution, and limit detection for each hormone need to be described.
- Only the information that was below 10% is not valid for this data.
- 2.7 Statistical analysis
Just one point of criticism in this study. Please improve the description of statistical analyses. In this way, we will be able to analyze whether ONE-WAY ANOVA is the best statistical test for the present study (I believe it is not the most appropriate). If the authors assume that they analyzed the results at different concentrations of MT, with the sex of the animals and with the time of exposure; ONE-WAY ANOVA is not indicated. Please review these reviews.
Results and Discussion sections
- Figures (4, 5, 6, and 7) are of low quality, and small size, which makes it difficult to see/reading.
- We know that the figures obtained by the KEGG software (for example) are not the most adequate, but when editing these images, a small improvement can be obtained.
- As mentioned above for the Introduction section. In the discussion long sentences were observed. Review the manuscript.
- Conclusion must be made based on the premises described at the final of introduction section. Conclusion strictly related to the results obtained. Depending on the assumptions/suggestions/probability/hypothesis, it can be considered in the discussion. Review the entire manuscript.
Reviewer 3 Report
Overall, the research manuscript is well-prepared. However, make minor revision as per the
comments provided in relevant sections.
Comment 1 (abstract section):
ï‚· Line no 4: G. rarus was exposed ………. Change to G. rarus were exposed
Comment 2 (Materials and methods):
2.1. Experimental animals
ï‚· How the doses of MT were selected for this study?
ï‚· Mention the sampling point in this section and why these time points were selected?
ï‚· Which types of tanks were used and mention their capacity? How many fish were kept in
each tank?
2.3. ELISA
ï‚· Were these assay kits validated for use in fish? Have they previously been reported in
fish research?
2.5. qRT-PCR
ï‚· Mention about primer designing protocol. Were they designed for this study or taken
from previous research paper?
ï‚· How the results were expressed? Was the 2 -ΔΔCT method used?
Comment 3 (Results):
Figure 1: Clarify male and female in each sub-figure? Different the significant levels in each
diagram. Use α symbol (17a-methyltestosterone) in figure caption.
Round 2
Reviewer 1 Report
Thank you for addressing the review comments. Good luck with your research!